# Machine Learning Model for Lymph Node Metastasis Prediction in Breast Cancer Using Random Forest Algorithm and Mitochondrial Metabolism Hub Genes

**Byung-Chul Kim [1], Jingyu Kim [2], Ilhan Lim [1], Dong Ho Kim [3], Sang Moo Lim [1] and Sang-Keun Woo [2,4,*]**

1   Department of Nuclear Medicine, Korea Institute of Radiological and Medical Sciences, Seoul 01812, Korea; xikian@kirams.re.kr (B.-C.K.); ilhan@kirams.re.kr (I.L.); smlim328@kirams.re.kr (S.M.L.)
2   Radiological & Medico-Oncological Sciences, University of Science & Technology, Seoul 34113, Korea; jingyu8754@kirams.re.kr
3   Department of Pediatrics, Korea Cancer Center Hospital, Seoul 01812, Korea; kdh@kirams.re.kr
4   Division of Applied RI, Korea Institute of Radiological and Medical Science, Seoul 01812, Korea
*   Correspondence: skwoo@kirams.re.kr; Tel.: +82-2-970-1659

**Abstract:** Breast cancer metastasis can have a fatal outcome, with the prediction of metastasis being critical for establishing effective treatment strategies. RNA-sequencing (RNA-seq) is a good tool for identifying genes that promote and support metastasis development. The hub gene analysis method is a bioinformatics method that can effectively analyze RNA sequencing results. This can be used to specify the set of genes most relevant to the function of the cell involved in metastasis. Herein, a new machine learning model based on RNA-seq data using the random forest algorithm and hub genes to estimate the accuracy of breast cancer metastasis prediction. Single-cell breast cancer samples (56 metastatic and 38 non-metastatic samples) were obtained from the Gene Expression Omnibus database, and the Weighted Gene Correlation Network Analysis package was used for the selection of gene modules and hub genes (function in mitochondrial metabolism). A machine learning prediction model using the hub gene set was devised and its accuracy was evaluated. A prediction model comprising 54-functional-gene modules and the hub gene set (*NDUFA9*, *NDUFB5*, and *NDUFB3*) showed an accuracy of $0.769 \pm 0.02$, $0.782 \pm 0.012$, and $0.945 \pm 0.016$, respectively. The test accuracy of the hub gene set was over 93% and that of the prediction model with random forest and hub genes was over 91%. A breast cancer metastasis dataset from The Cancer Genome Atlas was used for external validation, showing an accuracy of over 91%. The hub gene assay can be used to predict breast cancer metastasis by machine learning.

**Keywords:** sequence analysis; RNA/history; gene expression profiling/methods; models; statistical; algorithms; biomarkers; tumor



## 1. Introduction

RNA-sequencing (RNA-seq) is being used to diagnose cancer and predict the behavior of cancer cells [1], which is directly linked to the expression of certain genes. Thus, it is possible to diagnose breast cancer and predict metastasis by analyzing gene expression profiles [2]. Genes involved in metastasis can be identified by comparing RNA-seq results of confirmed metastatic and non-metastatic breast cancer samples. Genes such as *SETDB1* [3], *MALAT1* [4], *EHMT2* [5], *RAB11B-AS1* [6], *STAT3* [7], and *RAS* [8] were identified to play a role in lymph node metastasis of breast cancer. However, it is still impossible to effectively predict lymph node metastasis of breast cancer solely through gene expression analysis, although several studies have explored these particular genes. This limitation is because RNA-seq results only indicate the current state of breast cancer cells. Nevertheless, it is possible predict the behavior of breast cancer cells to some extent by analyzing the expression status of genes related to metastasis and grasping the current metastasis status of breast cancer [9], but the accuracy of such assessment is not high enough for clinical

application. Hub gene was used to obtain more accuracy to represent the function of cells than single gene analysis from RNS-sequencing in the cancer studies [10]. Hub gene was defined as genes having top 10% of connectivity in individual gene module related gene function [11]. The gene modules were created by a systematic biological strategy for evaluating gene association patterns among different samples with bioinformatics tools like WGCNA or GSEA [12].

Machine learning is a field of artificial intelligence that learns using algorithms based on real data, which is then used to design predictive models [13]. Such models are being actively investigated in different fields, including for the prediction of cancer prognosis using nuclear imaging [14]. However, prediction models building on nuclear imaging data still lack accuracy, making it difficult to apply clinically. This poor response of prediction models based only on imaging data can be explained as several factors can contribute to the cancer progression pattern. Recently, a prediction model with high accuracy was created by Dai et al. using genomic sequencing data of cancer cells that have metastasized from the colorectal cancer site to the lymph nodes [15]. This previous report demonstrates that it is possible to use next generation sequencing (NGS) data as input of prediction models with high accuracy.

In this study, a machine learning model based on RNA-seq data was developed using the random forest algorithm and hub genes to estimate the accuracy of the breast cancer metastasis prediction model. The accuracy of the prediction model with total genes, functional group genes, and hub genes was compared using various methods to identify a suitable machine learning algorithm for prediction of breast cancer metastasis.

## 2. Materials and Methods

### 2.1. Data Availability

Single-cell RNA-seq data deposited in the National Center for Biotechnology Information (NCBI) GEO database (Accession number: GSE75688), was used for data analysis. RNA-seq data of 94 single-cell samples from one double-positive ($ER^+$/$HER2^+$) breast cancer patient were used for this analysis. Illumina sequencing platform (HiSeq 2000, San Diego, CA, USA) was used to sequence cDNA libraries of 56 single-cell metastatic and 38 single-cell non-metastatic samples. Quality check was performed by fastQC (v.1.1.0).

### 2.2. Data Processing

Sequence data from patients were aligned to hg38 as a reference using STAR (version 2.7.0a) [16] with default parameters. Relative gene expression was determined by high-throughput sequencing (HTseq) [17]. To normalize gene expression levels, transcripts per million (TPM) (referenced by hg38) [18] were used. The selected data had a TPM value of greater than 1. To detect the differences in gene expression between metastatic and non-metastatic breast cancer, the DESeq2 package (version 1.24.0) in R was used [19]. Differentially expressed genes exceeding the cut off ($p$-value $< 0.05$ and $|$log2fold-change$| > 1$) were selected and their functions in metastasis were analyzed. Information on gene functions in metastasis was taken from the Human Cancer Metastasis Database (HCMDB) [20].

### 2.3. Principal Component Analysis (PCA) and Volcano Plot

For PCA, we filtered the data selecting TPM values greater than 1 from at least 10 samples. For the PCA plot, the prcomp function in "devtools" R package was used and ggbiplot package in R was used for visualization [21]. For visualization of the result of differentially expressed gene test, we used the ggplot package in R. In the volcano plot, the *x*-axis indicates the log2-fold change and *y*-axis indicates the *p*-value.

### 2.4. Weighted Gene Co-Expression Network Analysis

Weighted correlation network analysis (WGCNA) in R package was used to cluster genes and identify hub genes [22]. Before WGCNA, a matrix of unsigned Pearson correlations between all samples of transcripts was computed. The adjacency matrix, $a_{ij}$,

was calculated and the connection strength between each pair of nodes was determined using the following equation:

$$Sij = |cor(Xi, Xj)| aij = Sij\beta,$$

where Xi and Xj are vectors of the expression values for genes i and j, Sij represents Pearson's correlation coefficient of genes i and j, and aij indicates the network connection strength between genes i and j. We used the power at $\beta = 9$ (scale free R2 = 0.9) as a soft threshold parameter to ensure a scale-free network. In the co-expression network, genes with high absolute correlations were clustered into the same module. The WGCNA method considers the association between two connected genes as well as associated genes. Modules were also identified via hierarchical clustering of the weighting coefficient matrix. To further identify functional modules in the co-expression network with these 2863 genes, the adjacency matrix was transformed to a topological overlap matrix (TOM) to reduce noise and errors. The TOM was constructed using the following equation:

$$TOMi, j = \frac{\sum_{K=1}^{N} Ai, k \times Ak, j + Aij}{min(Ki, Kj) + 1 - Ai, j},$$

where A is the weighted adjacency matrix given by Aij = | cor(xi, × j)| $\beta$ and $\beta = 9$ is the soft threshold power. According to the TOM-based dissimilarity measure with a minimum size (gene group) of 30 for the gene dendrogram, average linkage hierarchical clustering was conducted, and genes with similar expression profiles were classified into one gene module using the DynamicTreeCut algorithm. To analyze average linkage hierarchical clusters and genes with similar expression patterns clustered together, 1-TOM was used. The expression profile of a given module was represented by its first principal component (module eigengene, MEs), which explained most of the variation in the module expression levels. The module eigengene-based connectivity of each gene was calculated by determining the correlation of the gene expression profile with the MEs. In addition, the gene significance (GS) was defined as the median p-value of each gene (GS = lgP) in the linear regression between gene expression and clinical traits. Module significance (MS) was defined as the average GS of all genes involved in the module and was measured to incorporate clinical information into the co-expression network.

*2.5. Hub Gene Identification and Validation*

The connectivity of genes was calculated by Pearson's correlation. Genes were considered as hub genes in the module when they showed high within-module connectivity (cor.geneModuleMembership; MM > 0.8). Hub genes in a given module tended to be strongly correlated with a certain clinical trait, which was measured as the absolute value of the Pearson's correlation (cor.geneTraitSignificance; GS > 0.8, *p*-value < 0.05). Hub gene clustering was visualized using the Contextual Hub Analysis Tool (CHAT) application in Cytoscape [23]. The functions of each gene set were investigated using the DAVID web tool (ver 6.8). Gene set with function in mitochondrial metabolism related to breast cancer metastasis [24] was selected for the prediction model. The hub gene processing was represented in diagram in Figure 1.

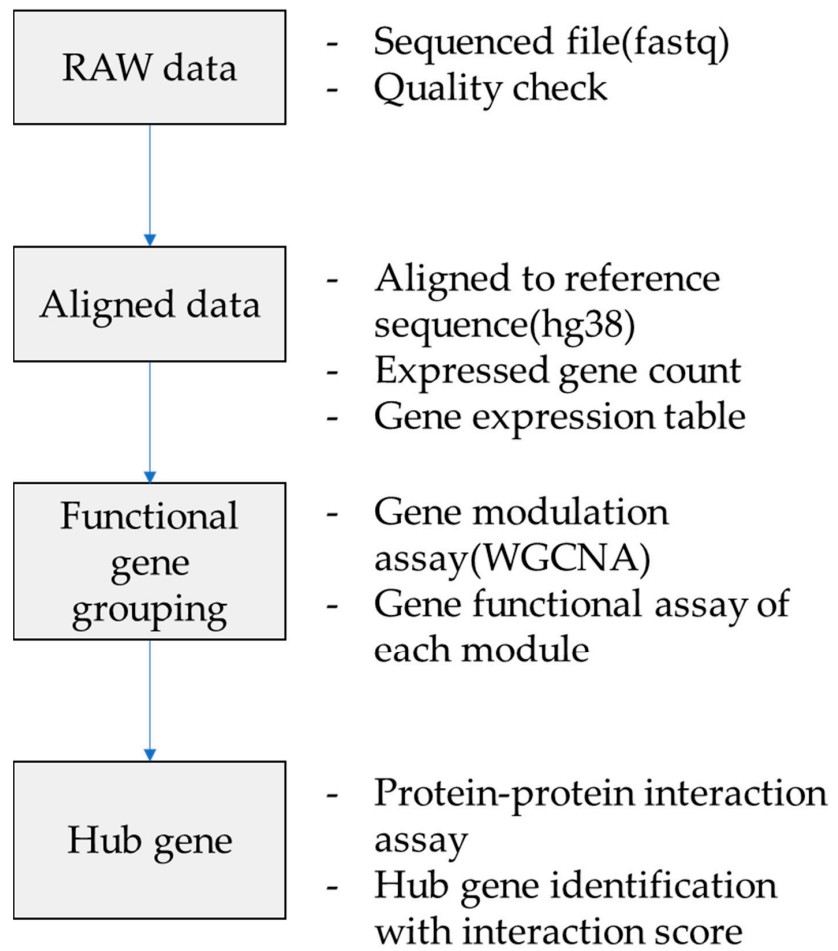

**Figure 1.** Diagram of hub gene identification process.

*2.6. Evaluation of the Metastasis Prediction Model*

To predict metastasis, we used machine learning techniques [25] such as logistic regression (LR) [26], linear discriminant analysis (LDA) [27], K-nearest neighbor algorithm (KNN) [28], classification and regression trees (CART) [29], naive Bayes classifier (NB) [30], support vector machine (SVM) [31], random forest (RF) [32], and gradient boost (GB) [33]. The machine learning prediction model was used to evaluate the accuracy, precision, and recall score using test data. An average of 10 predictions was taken as the final value [34]. The total gene expression model, selected module, and hub gene set were used for prediction using the machine learning algorithms. For external validation, RNA-seq data of 96 breast cancer samples, from The Cancer Genome Atlas (TGCA), were used. This data set consisted of 72 M0 macrophage non-metastatic and 24 M1 macrophage metastatic samples. To validate the prediction model, genes from the hub gene set was used in external validation with RF.

## 3. Results

*3.1. Difference between Breast Cancer and Metastatic Breast Cancer*

In this study, 94 single-cell samples from one breast cancer patient were used. Through RNA-seq analysis, 38 samples were defined as non-metastatic, and 56 samples as metastatic. These data were taken from the GEO database, which was developed by a previous breast cancer study that defined the cancer cell detected from lymph node of patients as metastatic cell, and the one detected from breast cancer as the non-metastatic one [31]. Low-quality reads (quality under 20) and short sequences (<35 base pairs) were trimmed. The distribution of each sample is displayed in PCA graph (Figure 1). After analysis of

differentially expressed genes and selection of genes as previously described, we identified 2683 genes showing significant differences between metastatic and non-metastatic samples. The results were visualized using a volcano plot (Figure 2). Gene expression in each sample type was visualized using pheatmap package in R (Figure 2). The top 20, most expressed genes (up/down regulated gene) are listed in Table 1.

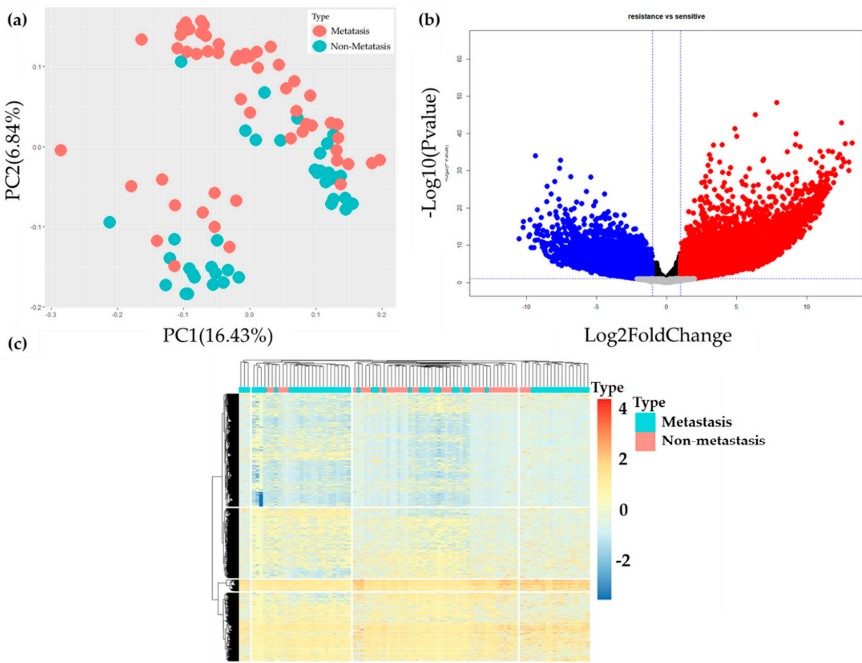

**Figure 2.** (**a**) PCA plot of samples showing distribution of each samples. (**b**) Volcano plot of differentially expressed genes. (**c**) Heatmap of gene expression in metastatic and non-metastatic samples.

**Table 1.** Fold change of genes by differentially expressed gene assay. The top-20 most up/down regulated genes by comparing non-metastatic breast cancer with metastatic breast cancer.

| Up-Regulated Genes | | | Down-Regulated Genes | | |
|---|---|---|---|---|---|
| Gene | logFC | *p*-Value | Gene | logFC | *p*-Value |
| IGLV3-19 | 14.31 | $7.2 \times 10^{-16}$ | IGLV2-14 | −14.10 | $1 \times 10^{-9}$ |
| RP1-298M8.1 | 12.98 | $1.3 \times 10^{-11}$ | IGLV1-44 | −10.86 | $2 \times 10^{-7}$ |
| IGHV3-48 | 12.56 | $6.5 \times 10^{-12}$ | IGHV4-31 | −10.75 | $2 \times 10^{-7}$ |
| RP11-39K24.12 | 12.50 | $6.6 \times 10^{-12}$ | IGLV1-40 | −10.25 | $2 \times 10^{-7}$ |
| ANXA1 | 12.31 | $1.1 \times 10^{-19}$ | IGLV1-47 | −9.93 | $7 \times 10^{-7}$ |
| RP11-39K24.13 | 12.03 | $1.6 \times 10^{-11}$ | IGKV2-28 | −9.92 | $3 \times 10^{-7}$ |
| RPL17P35 | 11.91 | $2.0 \times 10^{-12}$ | AIF1 | −9.76 | $5 \times 10^{-7}$ |
| RN7SL318P | 11.81 | $9.6 \times 10^{-11}$ | IGLV7-46 | −9.68 | $1 \times 10^{-6}$ |
| RNU1-8P | 11.74 | $6.4 \times 10^{-12}$ | IGKV1D-39 | −9.32 | $4 \times 10^{-6}$ |
| OR10G3 | 11.73 | $1.8 \times 10^{-12}$ | SNORA70G | −9.17 | $6 \times 10^{-7}$ |
| RP11-83M16.2 | 11.62 | $1.2 \times 10^{-10}$ | IGHV3-30 | −9.16 | $8 \times 10^{-6}$ |
| AC007560.1 | 11.54 | $2.9 \times 10^{-9}$ | RN7SL237P | −8.97 | $2 \times 10^{-4}$ |
| FCGR2A | 11.47 | $8.7 \times 10^{-11}$ | RP11-90O23.1 | −8.89 | $2 \times 10^{-4}$ |
| RN7SL321P | 11.46 | $1.9 \times 10^{-10}$ | TRBV5-6 | −8.82 | $2 \times 10^{-4}$ |
| RNU4ATAC16P | 11.46 | $1.9 \times 10^{-10}$ | RP11-571F15.3 | −8.81 | $2 \times 10^{-5}$ |
| KRT19P2 | 11.42 | $7.1 \times 10^{-11}$ | RP11-613F22.5 | −8.79 | $5 \times 10^{-5}$ |
| BRS3 | 11.33 | $1.3 \times 10^{-11}$ | IGKV1-39 | −8.77 | $2 \times 10^{-4}$ |
| PSMA6P4 | 11.27 | $8.4 \times 10^{-11}$ | IGLV10-54 | −8.67 | $2 \times 10^{-4}$ |
| RP11-64K12.1 | 11.23 | $3.6 \times 10^{-10}$ | IGKV1-5 | −8.66 | $1 \times 10^{-4}$ |
| CYCSP39 | 11.22 | $3.7 \times 10^{-10}$ | RP11-389C8.3 | −8.56 | $3 \times 10^{-4}$ |

### 3.2. Weighted Co-Expression Network Construction and Key Module Identification

Co-expression analysis was carried out to construct a co-expression network (Figure 3). Soft-thresholding power in WGCNA was determined for clustering gene expression

(Figure 4). We found 2147 genes of genes of most variance genes (MVGs) with similar gene expression patterns. These genes were included in the modules following cluster analysis. Eight modules showing the same gene expression patterns were identified and analyzed, two of which were related to the metastatic stage and the other six modules were related to the non-metastatic stage. We visualized the gene network with a heatmap and meta-modules (Figure 5). Each colored module represents a gene group with similar gene expression patterns, calculated by WGCNA function. We selected the brown module for analysis, which was the most closely associated with the metastatic stage; this module was crucial to evaluating breast cancer metastasis (R = 0.39, *p*-value = $1 \times 10^{-4}$).

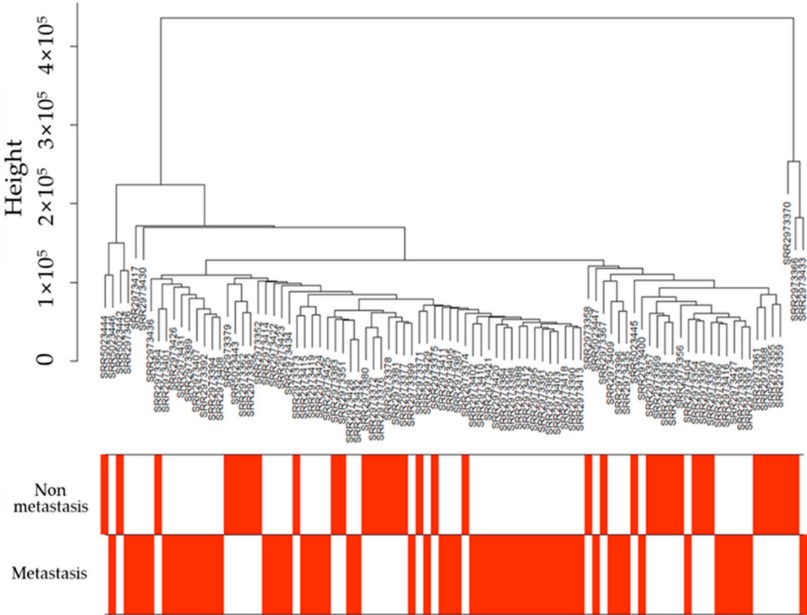

**Figure 3.** Clustering dendrogram of 94 samples.

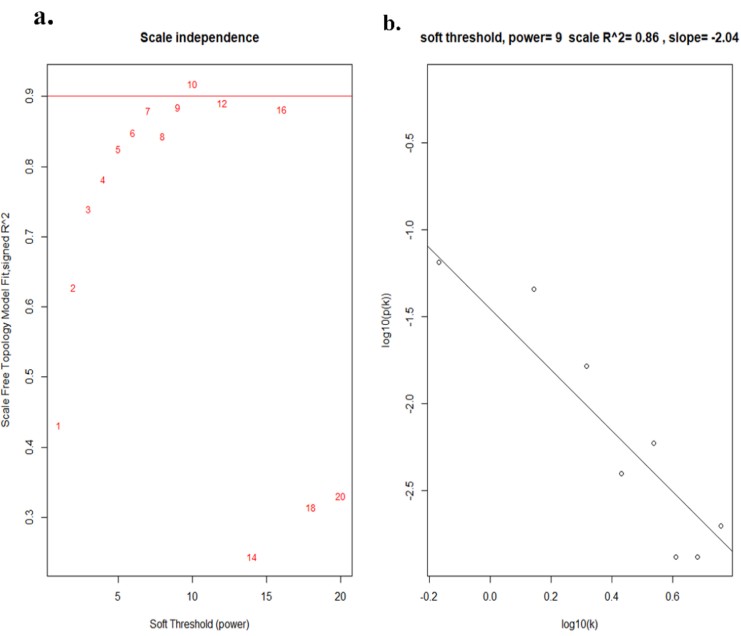

**Figure 4.** Determination of soft-thresholding power in WGCNA. (**a**) Analysis of the scale-free fit index for various soft-thresholding powers (β). (**b**) Analysis of the mean connectivity for various soft-thresholding powers. (**b**) Evaluating the scale free topology when β = 9.

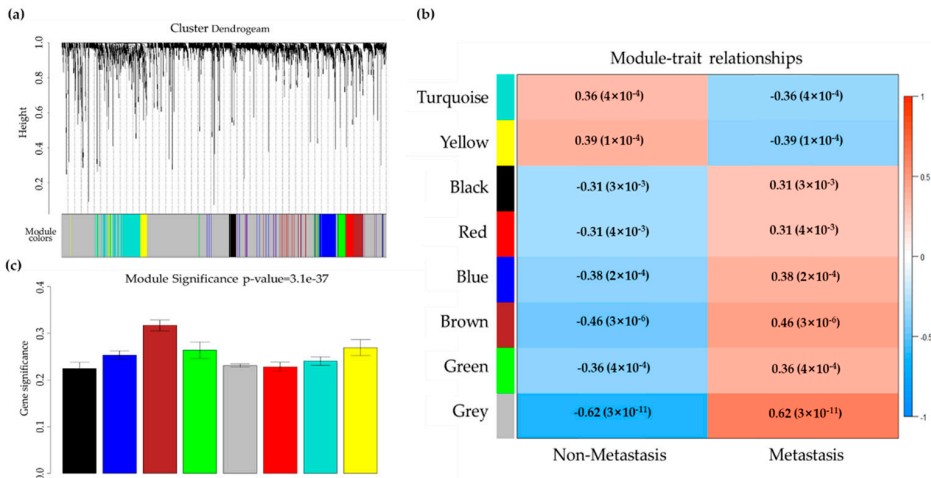

**Figure 5.** Identification of modules associated with the clinical traits of breast cancer. (**a**) Dendrogram of all differentially expressed genes clustered based on a dissimilarity measure (1-TOM). Eight modules were identified by WGCNA assay. (**b**) Heatmap of the correlation between module eigengenes and clinical traits of breast cancer. Two modules have non-metastasis related gene expression pattern and rest 6 modules have metastasis related gene expression pattern. (**c**) Distribution of average gene significance and errors in the modules associated with metastasis and non-metastasis breast cancer. The brown module was selected for assay because it had the highest gene significance in metastasis statement.

### 3.3. Gene Ontology (GO) and Pathway Enrichment Analysis

We performed functional genes assay of genes from the brown module that have significant metastatic activity. The identified genes were categorized into three functional groups: biological process, cellular component, and molecular function (BP, CC, and MF, respectively). In the BP group, mRNA catabolic process, translation initiation, nuclear-transcribed mRNA catabolic process, nonsense-mediated decay, protein targeting to membrane, protein targeting to ER, establishment of protein localization to endoplasmic reticulum, protein localization to endoplasmic reticulum, SRP-dependent co-translational protein targeting to membrane, and co-translational protein targeting to membrane were mainly enhanced; in the CC group, ribosome, ribosomal subunit, cytosolic part, cytosolic ribosome, focal adhesion, cell-substrate adhesion junctions, apical junction complex, lateral plasma membrane, and apicultural plasma membrane increased; and in the MF group, structural constituent of ribosome, cell adhesion molecule binding, cadherin binding, ubiquitin protein ligase binding, unfolded protein binding, ribonucleoprotein complex binding, rRNA binding, threonine-type endopeptidase activity, threonine-type peptidase activity, and mRNA 5′-UTR binding were enhanced. According to Kyoto Encyclopedia of Genes and Genomes (KEGG) pathway analysis, our results demonstrated that these genes were mainly involved in ribosome, spliceosome, proteasome, and Parkinson disease. These results indicate that the genes from the brown module are involved in cell proliferation and cell migration (Figure 6).

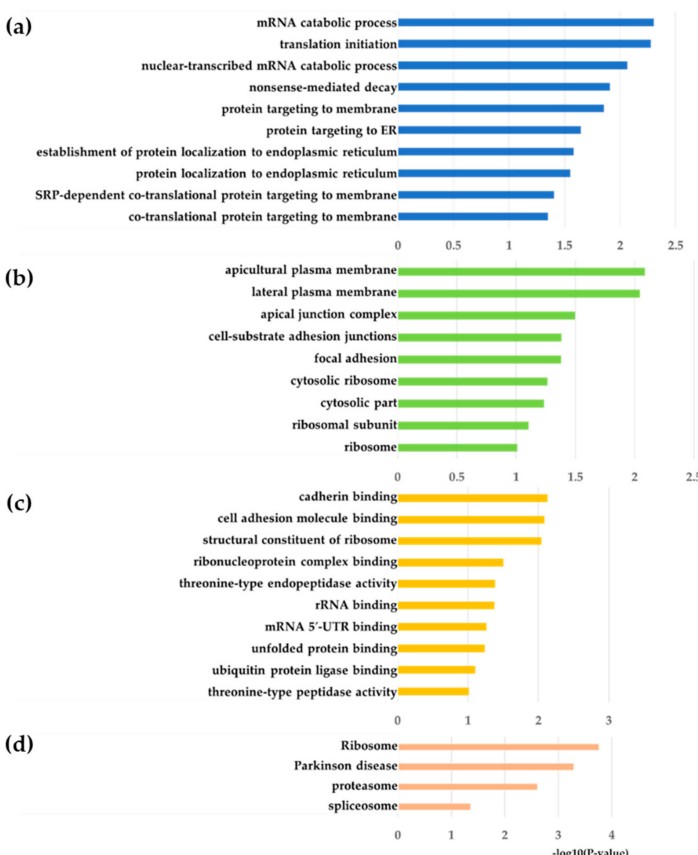

**Figure 6.** Gene ontology and pathway enrichment analysis of blue module genes. *X*-axis reveal the −log10 (*p*-value). (**a**) Cellular component analysis. (**b**) Biological process analysis. (**c**) Molecular function analysis. (**d**) KEGG pathway analysis.

### 3.4. Estimation of the Prediction Model

Total expression genes (16,482 genes), selected modules (54 genes), and hub gene set (3 genes) were used for evaluating the prediction model of metastasis. The hub gene set consisted of 3 genes: *NDUFA9*, *NDUFB5*, and *NDUFB3* (Figure 7). RF algorithm was selected for the prediction model because it had highest test accuracy with the hub gene set. The accuracy of the total genes test, the functional gene modules, and for the hub gene set test was highest with RF (0.769, 0.782, and 0.945, respectively). The train accuracy and test accuracy were both found to be 0.910, by external validation with breast cancer patient data from TCGA and RF (Table 2).

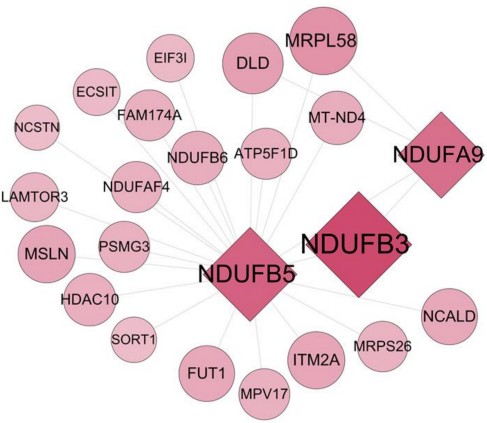

**Figure 7.** Hub genes (in squares) that showed the strongest relationship with mitochondrial metabolism, and other genes that interact with the hub genes (in circles).

**Table 2.** RNA-sequencing data of accuracy of various machine learning techniques in breast cancer result. Test accuracy of each algorithm using total gene, functional gene module, and hub gene set (3).

| Machine Learning Algorithm | Total Gene (16,482) | | Functional Gene Module (54) | | Hub Gene Set (3) | | External Validation (3) | |
|---|---|---|---|---|---|---|---|---|
| | Test | Train | Test | Train | Test | Train | Test | Train |
| LR | 0.571 | 0.571 | 0.429 | 0.593 | 0.714 | 0.587 | 0.759 | 0.746 |
| LDA | 0.429 | 0.429 | 0.429 | 0.614 | 0.522 | 0.603 | 0.759 | 0.746 |
| KNN | 0.429 | 0.831 | 0.571 | 0.714 | 0.714 | 0.757 | 0.667 | 0.825 |
| CART | 0.457 | 0.55 | 0.528 | 0.592 | 0.761 | 0.575 | 0.731 | 0.627 |
| NB | 0.571 | 0.571 | 0.571 | 0.582 | 0.571 | 0.566 | 0.655 | 0.642 |
| SVM | 0.841 | 0.841 | 0.788 | 0.661 | 0.571 | 0.571 | 0.746 | 0.746 |
| RF | 0.769 | 0.769 | 0.782 | 0.866 | 0.945 | 0.945 | 0.910 | 0.910 |
| GB | 0.758 | 0.758 | 0.844 | 0.852 | 0.857 | 0.852 | 0.897 | 0.881 |

## 4. Discussion

In this study, we applied the hub gene set from RNA-seq and machine learning algorithms for the first time to construct predictive models for breast cancer metastasis with high accuracy, contrasting with most of the previous studies that have focused on nuclear imaging data. Although the results of analyzing imaging factors, such as computerized tomography and magnetic resonance imaging were applied to machine learning to estimate the accuracy of the predictive model, it is still difficult to apply in the clinical setting. Genetic data obtained by next generation sequencing can describe the patient clinical condition more accurately than data obtained solely by image analysis. Therefore, we expected that in the construction of this prediction model, higher accuracy than the existing image analysis-based prediction model could be estimated.

RNA-sequencing techniques and machine learning algorithms was used to build a model to predict breast cancer metastasis, which is promoted by genetic changes within cancer cells. Such a model can be used as the basis for diagnosis and treatment of breast cancer by analyzing changes in genetic events. In the past, only a limited number of genes could be analyzed to profile cancer cells, but advances in technology have enhanced the analytical methods, allowing the assessment of the genetic landscape of cells at once. RNA-seq is believed to hold the power to explain the behavior of cancer by identifying key genes. However, the behavior of cancer cells is difficult to explain based solely on the gene expression of one gene. To overcome this limitation, herein we used the concept of a functional group gene and a hub gene. The expression of a hub gene can accurately determine the function of a cell rather than using a single gene since it is closely related to the function of the cell. Several genes were found to be involved when the cancer cell function was addressed. Hence, genes involved in the same function were defined as the functional group genes, among which genes with the most essential functions were selected and defined as the hub genes.

WGCNA, a bioinformatics analysis method, was used to identify the functional group gene and the hub gene [35]. This approach allows to evaluate the expression pattern of a gene and construct a module of genes with similar expression patterns, which is generally used for gene analysis of cancer cells [36]. This method has the advantage of being able to search for a function related to a highly expressed gene among the functions of a cell, as it analyzes only the expression pattern of the gene. Gene modules were defined as functional groups, and hub genes were retrieved from within. The hub gene was selected as the gene with the most connections after confirming the association between the genes in the analyzed module using CHAT app in Cytoscape [37]. Therefore, the identified functional group and three hub genes could represent a more detailed and reliable indicator of lymph node metastasis in breast cancer.

A detailed analysis of the functional group genes revealed that they were mainly involved in cell division and proliferation (Figure 6). Epithelial mesenchymal transition (EMT) is the most frequently expressed function in tumor metastases, and cell division and

proliferation are active and essential actions during EMT and cancer cells spreading [38]. The identified hub genes (*NDUFB3, NDUFB5,* and *NDUFA9*) are mitochondrial genes that have various functions, but are known to be involved in mitochondria metabolism. This function is known to play an important role in metastasis of cancer cells and is reported to be overexpressed when metastasis of breast, rectal, and head and neck cancers occurs [39–41].

We used a machine learning method to construct a highly accurate predictive model for breast cancer metastasis. Several algorithms could be used for the machine learning process; hence, since it was not known which algorithm was suitable for breast cancer metastasis, we tested various algorithms. Among the algorithms used to estimate the predictive model, RF and GB showed the highest accuracy. These two most suitable algorithms revealed the highest degree of analysis because the number of analyzed samples was small, and a prediction model was constructed by extracting the most suitable few factors for analysis [42,43]. The sample used to design the predictive model was not general result since it was based on single cell RNA-seq data. Therefore, a large-scale breast cancer metastasis TCGA RNA-seq dataset was used for external validation, which was classified into the M0 or M1 group with reference to the clinical data. The breast cancer metastasis model using RF showed a high accuracy of 94%, and the external validation also showed a high accuracy of 91%.

## 5. Conclusions

In this study, we applied RNA-seq data and machine learning algorithms for the first time to construct a highly accurate predictive model to predict breast cancer metastasis. The field of predicting patient prognosis using machine learning has been mainly applied to nuclear imaging. Although the results of analyzing imaging factors, such as computerized tomography and magnetic resonance imaging, were applied to machine learning to estimate the accuracy of the predictive model, it is still difficult to apply such approaches in the clinical setting. Genetic data obtained by RNA sequencing can describe the patient condition more accurately than data obtained by image analysis. Therefore, we expect that the novel predictive model may pave the way for enhance prognosis assessment of patients with breast cancer. Additionally, as we combined RNA-seq data with machine learning to estimate predictive models with high levels of accuracy, applying nuclear imaging and RNA-seq to predictive models at the same time will allows us to estimate better predictive models. It is expected that the predictive model estimated through this combined approach may be used in clinical practice.

**Author Contributions:** B.-C.K.; conceptualization, B.-C.K. and S.-K.W.; methodology, B.-C.K.; software, B.-C.K.; validation, B.-C.K. and J.K.; formal analysis, B.-C.K.; investigation, B.-C.K.; data curation, B.-C.K.; writing—original draft preparation, B.-C.K.; writing—review and editing, I.L., D.H.K., S.M.L. and S.-K.W.; visualization, B.-C.K.; supervision, S.-K.W.; project administration, I.L. All authors have read and agreed to the published version of the manuscript.

**Funding:** This study was supported by the National Research Foundation of Korea (NRF) grant funded by the Korea government (Ministry of Science and ICT) (No. 2020M2D9A1094070).

**Institutional Review Board Statement:** This study was approved by the institutional review board (IRB) (e-IRB number: KIRAMS 2021-02-003).

**Informed Consent Statement:** Not applicable.

**Data Availability Statement:** Not applicable.

**Acknowledgments:** Not applicable.

**Conflicts of Interest:** The authors declare no conflict of interest.

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
