# Peer review of "Machine Learning Model for Lymph Node Metastasis Prediction in Breast Cancer Using Random Forest Algorithm and Mitochondrial Metabolism Hub Genes"

_applsci, doi:10.3390/app11072897_

Round 1

Reviewer 1 Report

This paper focuses on the machine learning model for breast cancer metastasis prediction by using random forest algorithm and mitochondrial metabolism hub gene assay.

Line 50 – put the capital I in the beginning of the sentence - it is possible…

Line 51 – put the t – letter in front of the word – he RAS

Line 109 – put the capital s in the beginning of the Pearson’s correlation coefficient – sij

Line 144 – put the capital G in the beginning of the word –gene set….

Line 159 – put the capital R at the following word – 3.1. result

Line 172 – put the capital in front of the text – the top…

Line 218 – Figure 1 a) and b) are not so clear, please increase the resolution.

Line 223 – Figure 3 a) , b) and c) are not so clear, please increase the resolution.

Line 226 – Figure 4 a) , b) , c) and d) are not so clear, please increase the resolution.

Line 240 – remove one m from the word – metabolism

Line 268 – put number 3 instead of number 4  at the following text – Fig 3

Line 279 – put the capital N in front of the word Nor-

Line 282 – put the capital C in front of the word Cell

Line 283 – remove a full stop

The structure of the - testing various machine learning algorithms is missing. Please do the design of it.

More diagrams should be included in the paper, to have clear picture about the obtained results, for example 3.1.3. GO and pathway enrichment analysis is missing such a diagram.

Author Response

January 26, 2021

Dr. Joy Sun

Assigned Editor

Applied Sciences

Dear Dr. Joy Sun:

We would like to resubmit manuscript applsci-1079120, title Machine learning model for breast cancer metastasis prediction using random forest algorithm and mitochondrial metabolism hub gene assay for publication in Applied Sciences.

We thank you and the reviewers for your kind and helpful comments. We have revised the manuscript accordingly. Point-by-point responses to each comment are listed below.

We hope that this revised manuscript is now suitable for publication in your journal. Thank you for your consideration.

Sincerely,

Sang-Keun Woo, Ph.D.

Department of Nuclear Medicine,

KIRAMS, 75 Nowon-gu, Nowon-ro, Seoul, Republic of Korea

E-mail: skwoo@kirams.re.kr

--------------------------------------------------------------------------------------------------------------

Reviewers' comments:

Reviewer #1: This paper focuses on the machine learning model for breast cancer metastasis prediction by using random forest algorithm and mitochondrial metabolism hub gene assay.

Line 50

put the capital I in the beginning of the sentence

Response: We have made the necessary change.

Line 109

put the capital S in the beginning of the Pearson’s correlation coefficient –

Response: We have changed “sij” to “Sij”.

Line 144

put the capital G in the beginning of the word

Response: We have changed “gene set” to “Gene set”.

Line 159

put the capital R at the following word

Response: We have changed “3.1. result” to “3.1. Result”.

Line 172

put the capital in front of the text

Response: We changed “the top” to “The top”.

Line 218

Figure 1 a) and b) are not so clear, please increase the resolution.

Response: We have increased the resolution of the image.

Line 223

Figure 3 a), b) and c) are not so clear, please increase the resolution.

Response: We have increased the resolution of the image.

Line 226

Figure 4 a) , b) , c) and d) are not so clear, please increase the resolution.

Response: We have increased the resolution of the image.

Line 240

remove one m from the word

Response: We have changed “metabolismm” to “metabolism”.

Line 268

put number 3 instead of number 4 at the following text

Response: We changed “Fig 3” to “Fig 4”.

Line 279

put the capital N in front of the word Nor-

Response: We have changed “normally” to “Normally”.

Line 282

put the capital C in front of the word Cell

Response: We have changed “cell” to “Cell”.

Line 283

remove a full stop

Response: We have changed “..” to “.”

Reviewer 2 Report

This study is dealing with a very interesting topic in medical application of machine learning algorithm analysis. However, the manuscript is poorly prepared to present the content of the study properly. There should be an extensive rewriting of manuscript according to the points as suggested below. The English and the grammar need to be checked thoroughly by native speakers or commercial editing services.

1) Title

The title doesn't seem to present the content of the study properly. I suggest you to modify as below or so.  

Machine learning model for lymph node metastasis prediction in breast cancer using random forest algorithm and Gene Expression Omnibus RNA-seq database of mitochondrial metabolism hub genes

2) Abstract

Abstract is not written structurally so it is not easy to catch what this study is trying to find out and how this study was designed and what was the results. i understand this study is a huge project and includes lots of analysis so it is hard to explain everything in the limited space. Thus structure of paragraph is very important to make readers understand this study properly. 

The first sentence of abstract is recommended to include the general background relevant to this study. It should be well summarized as well as one sentence such as below.

"Recently, machine-learning models based on DNA/RNA data are being developed and showing a promising results on predicting cancer behavior and prognosis."

The second sentence of abstract should include current problems or situation, obstacles, or issues that should be overcome to address why this study is important or why this study is needed. For example, as below.

"Recently, researchers have tried to find the key genetic changes that can be used to predict the metastasis in breast cancers using whole genome/transcriptome data. However, because of the magnitude of these data, it is not easy to determine the most effective genes for metastatic marker among thousands of differently expressed genes. In this study, we developed a machine-learning model based on RNA-seq data using random forest algorithm to find the key metastatic markers."

Material methods part in this abstract is written as not easy to understand. It should be re-written to be understood intuitively. How many data (or samples) was used. The full terms for GEO, WGCNA, NGS, HUB should be addressed at their first appearance if you want to use these terms.

Results and methods are admixed at present abstract. It is better to separate and make sure what is methods and results. The content of the results should be corresponded to somethings described in methods.  

Conclusion should be corresponded to the purpose of this study. 

It is recommended to use MeSH terms for Keywords 

3) Introduction

It is recommended to write introduction as 3 paragraphs. First sentences of each paragraph should be a topic sentence that represent the content of each paragraph like a summary sentence so that readers can understand easily what authors are trying to say only by reading the first sentences of introduction. It is recommended to corresponded to the first three sentences of abstracts. 

Do not try to use too much space for explaining the basic concepts and background knowledge. Please remember this is not review paper. 

Three paragraphs should explain why this study is important (background), what is current situation and problems (recent study findings), how this study can solve this problem (the purpose and the design of this study).

4) Materials and Methods/Results

there is no IRB statement. Even though this study are using open data, the waiver of IRB should be taken by IRB and it should be mentioned.

row 76: "RNA-seq (single-cell) data have been deposited in the NCBI Gene Expression Omnibus database."

This sounded like you have deposited gene data in NCBI database.

"RNA-seq (single-cell) data deposited in the NCBI Gene Expression Omnibus database was downloaded and used for data analysis." or

"We used RNA-seq (single-cell) data deposited in the NCBI Gene Expression Omnibus database to find the key RNA genes that is related to metastasis.

is less confusing. Every other part of methods should be revised likewise.

Many process during the analysis are described in results part not in methods which makes readers very confusing. and many results are included in methods parts. All the analytic methods should be described in methods parts and All the results that is should be described in results. subheadings of methods and results should be corresponded. for example, in 2.1 data collection of methods, you need to describe the methods how you collected the data. And in results, under the subheading of Enrolled data, you need to describe how many data have been finally collected for the analysis. Likewise in 3.1.1. (row164), the part that you are explaining about the definition of non-metastatic and metastatic should be placed in methods parts than results part. row 166 "The quality of sequencing data was 166
checked using FastQC tools [32], and PCA analysis [33] was performed to determine the distribution of each sample" this is also methods not results.  

5) Discussion

The first paragraph should state the findings of this study which is the answer for hypothesis of introduction, the solution for problems of introduction.

The first sentence of first paragraph should mention the most important findings or novel findings. for example,

"In this study, we found 1., 2., 3., ...."

"In this study, we verified 1, 2, 3,..."

and you should explain these findings one by one from the next paragraphs.

"We have proved finding #1 with ...  this analysis...."

and you can mention some previous findings from other relevant studies.

and address the importance of your findings. the difference of your study than other studies. or novel finding of your study. etc.

The length of discussion is too short and the discussion should be separated at least 4-5 paragraphs according to the topics. 

The limitation of this study should be discussed. I think this study needs external validation using real patient data or other database data. because single cell RNA-seq data from 'only one' patient have been used in this study. It is impossible to generalize the findings of this study.

Author information

One author's affiliation (dept of pediatrics) seems to be not relevant for this study. This study is not for pediatric cases.

Funding

funding information is misplaced in acknowledgment although this article has a funding information section separately.

References

There are many errors on the references, wrong use of upper and lower cases and inclusion of % by mistakes. Please check carefully once again on the references.

Author Response

January 26, 2021

Dr. Joy Sun

Assigned Editor

Applied Sciences

Dear Dr. Joy Sun:

We would like to resubmit manuscript applsci-1079120, title Machine learning model for breast cancer metastasis prediction using random forest algorithm and mitochondrial metabolism hub gene assay for publication in Applied Sciences.

We thank you and the reviewers for your kind and helpful comments. We have revised the manuscript accordingly. Point-by-point responses to each comment are listed below.

We hope that this revised manuscript is now suitable for publication in your journal. Thank you for your consideration.

Sincerely,

Sang-Keun Woo, Ph.D.

Department of Nuclear Medicine,

KIRAMS, 75 Nowon-gu, Nowon-ro, Seoul, Republic of Korea

E-mail: skwoo@kirams.re.kr

--------------------------------------------------------------------------------------------------------------

Reviewers' comments:

Reviewer #2: This study is dealing with a very interesting topic in medical application of machine learning algorithm analysis. However, the manuscript is poorly prepared to present the content of the study properly. There should be an extensive rewriting of manuscript according to the points as suggested below. The English and the grammar need to be checked thoroughly by native speakers or commercial editing services.

1) Title

The title doesn't seem to present the content of the study properly. I suggest you to modify as below or so. 

- Machine learning model for breast cancer metastasis prediction using random forest algorithm and mitochondrial metabolism hub gene assay

Response: Thank you for your kind comment on the title, and we accept your suggestion. However, the “Gene Expression Omnibus RNA-seq database of is expected in the title because we used only the data on breast cancer with metastasis from GEO database and not the entire GEO database. We changed the title as follows:

Machine learning model for lymph node metastasis prediction in breast cancer using random forest algorithm and mitochondrial metabolism hub genes

2) Abstract

Abstract is not written structurally so it is not easy to catch what this study is trying to find out and how this study was designed and what was the results. I understand this study is a huge project and includes lots of analysis so it is hard to explain everything in the limited space. Thus, structure of paragraph is very important to make readers understand this study properly.

- The first sentence of abstract is recommended to include the general background relevant to this study. It should be well summarized as well as one sentence such as below.

The machine learning prediction model created based on the results of RNA-sequencing using breast cancer metastasis tissue can predict the metastasis potential of patients

Response: Thank you for the kind comment. The first sentence of the abstract has been changed.

Machine-learning models that successfully predict breast cancer behavior and prognosis using genomic data are being developed.

- The second sentence of abstract should include current problems or situation, obstacles, or issues that should be overcome to address why this study is important or why this study is needed. For example, as below.

- In this study, we estimated the optimal predictive model for breast cancer metastasis using hub gene analysis.

Response: Thank you for your kind comment. The second sentence of the abstract has also been changed.

Researchers are trying to find gene mutations that can be used to predict metastasis in breast cancer using whole-genome or transcriptome data. However, it is difficult to determine the best metastatic marker genes among thousands of differentially expressed genes. In this study, we have developed a machine learning model based on RNA-sequencing (RNA-seq) data using random forest algorithm and hub genes to estimate the accuracy of the breast cancer metastasis prediction model.

- Material methods part in this abstract is written as not easy to understand. It should be re-written to be understood intuitively. How many data (or samples) was used. The full terms for GEO, WGCNA, NGS, HUB should be addressed at their first appearance if you want to use these terms.

Response: Thank you for your kind comment. The number of patients used in the analysis is mentioned and the full forms of abbreviations used have been added.

RNA-seq data, for breast cancer metastasis, were obtained from the Gene Expression Omnibus (GEO) database. A total of 94 single-cell breast cancer samples (56 metastatic and 38 non-metastatic samples) were used for the RNA-seq analysis. Weighted Gene Correlation Network Analysis (WGCNA) package was used for the selection of gene modules and hub gene set (function in mitochondrial metabolism).

- Results and methods are admixed at present abstract. It is better to separate and make sure what is methods and results. The content of the results should be corresponded to somethings described in methods. 

Response: Thank you for your kind comment. In the abstract, the method and result are clearly divided and described as advised.

A total of 94 single-cell breast cancer samples (56 metastatic and 38 non-metastatic samples) were used for the RNA-seq analysis. Weighted Gene Correlation Network Analysis (WGCNA) package was used for the selection of gene modules and hub gene set (function in mitochondrial metabolism). A machine learning prediction model using the hub gene set was devised and its accuracy was evaluated. Prediction model with total gene, 54 functional gene module, and hub gene set were compared. The accuracy of machine learning model with total gene, 54 functional gene module and 3 mitochondrial metabolism hub genes (NDUFA9, NDUFB5, NDUFB3) was 0.769 ± 0.02, 0.782 ±0.012, and 0.945 ± 0.016, respectively. The test accuracy of the hub gene set was over 93%.

Conclusion should be corresponded to the purpose of this study.

Response: Thank you for your kind comment. We have changed the sentence as follows:

As a result of testing various machine learning algorithms, we found that RF and GB have the highest accuracy (over 85%) in predicting breast cancer metastatic models. In addition, the highest accuracy was obtained when the hub gene set was used as an input. Therefore, the predictive machine learning model using the three hub genes, NDUFA9, NDUFB5, and NDUFB3, showed a possibility of using a hub gene to predict breast cancer metastasis.

It is recommended to use MeSH terms for Keywords

Response: Thank you for your kind comment. We have changed the keywords to the MeSH format “1; RNA-sequencing 2; hub gene 3; prediction model 4; machine learning 5; breast cancer metastasis” to “1; Sequence Analysis, RNA / history 2; Gene Expression Profiling / methods 3; Models, Statistical 4; Algorithms 5; Biomarkers, Tumor”

3) Introduction

It is recommended to write introduction as 3 paragraphs. First sentences of each paragraph should be a topic sentence that represent the content of each paragraph like a summary sentence so that readers can understand easily what authors are trying to say only by reading the first sentences of introduction. It is recommended to corresponded to the first three sentences of abstracts.

Do not try to use too much space for explaining the basic concepts and background knowledge. Please remember this is not review paper.

Three paragraphs should explain why this study is important (background), what is current situation and problems (recent study findings), how this study can solve this problem (the purpose and the design of this study).

Response: Thank you for your kind comment. Introduction has been re-written in 3 paragraphs as per your advice. It consists of background, recent study findings, and the purpose and design of this study. The re-written Introduction is as described below; 

RNA-sequencing (RNA-seq) is being used to diagnose cancer and predict the behavior of cancer cells [2]. The behavior of cancer cells is directly linked to the expression of certain genes. Thus, it is possible to diagnose breast cancer and predict metastasis by analyzing gene expression [3]. Genes involved in metastasis can be identified by comparing the results of RNA-seq analysis of breast cancer tissues with confirmed metastatic and non-metastatic samples. Genes such as SETDB1 [4], MALAT1 [5], EHMT2 [6], RAB11B-AS1 [7], STAT3 [8], and RAS [9], have been identified to play a role metastasis. It is difficult to accurately analyze the function of a cell by analyzing the expression level of an individual gene.

Machine learning is a field of Artificial Intelligence that learns using algorithms based on data and makes predictive models using the learned results [11]. It is widely used due to its accuracy in predicting models using large amount of data that is difficult for humans to analyze. Many studies have predicted cancer behavior using machine learning. Recently, a prediction model was created using the results of genomic sequencing, for the prognosis of cancer treatment. Dai et al. showed a predictive model using the transcriptomic data of cancer cells that have metastasized from colorectal cancer to lymph nodes [12].

In this study, we try to find a machine learning algorithm suitable for the prediction of breast cancer metastasis, and evaluate the accuracy of the prediction model using hub gene set. We compared the accuracy of the prediction model with total genes, functional group genes, and hub genes using various machine learning algorithms.

4) Materials and Methods/Results

there is no IRB statement. Even though this study is using open data, the waiver of IRB should be taken by IRB and it should be mentioned.

Response: Thank you for your kind comment. We have added an IRB statement in the Methods section.

The IRB approval was waived because the data used for analysis was open data.

row 76:

RNA-seq (single-cell) data have been deposited in the NCBI Gene Expression Omnibus database."

This sounded like you have deposited gene data in NCBI database.

Response: Thank you for your kind comment. Sentences that are ambiguous have been changed.

Single-cell RNA-seq data deposited in the National Center for Biotechnology Information (NCBI) GEO database (Accession number: GSE75688), was used for data analysis.

row 166

"The quality of sequencing data was 166 checked using FastQC tools [32], and PCA analysis [33] was performed to determine the distribution of each sample" this is also methods not results.

Response: Thank you for your kind comment. The sentence has been replaced.

The distribution on each sample is displayed in PCA graph (Fig 1).

Many processes during the analysis are described in results part not in methods which makes readers very confusing. and many results are included in methods parts. All the analytic methods should be described in methods parts and All the results that is should be described in results. subheadings of methods and results should be corresponded. for example, in 2.1 data collection of methods, you need to describe the methods how you collected the data. And in results, under the subheading of Enrolled data, you need to describe how many data have been finally collected for the analysis

A total of 563 samples were collected from 13 patients.

Response: Thank you for your kind comment. Ambiguous sentences have been corrected and the number of samples used in the actual experiment is mentioned.

96 single cell RNA-sequencing data from one person were used for this analysis.

5) Discussion

The first paragraph should state the findings of this study which is the answer for hypothesis of introduction, the solution for problems of introduction.

The first sentence of first paragraph should mention the most important findings or novel findings. for example,

"In this study, we found 1., 2., 3., ...."

"In this study, we verified 1, 2, 3,..."

and you should explain these findings one by one from the next paragraphs.

"We have proved finding #1 with ...  this analysis...."

and you can mention some previous findings from other relevant studies.

and address the importance of your findings. the difference of your study than other studies. or novel finding of your study. etc.

The length of discussion is too short and the discussion should be separated at least 4-5 paragraphs according to the topics.

Response: Thank you for your kind comment. We have rewritten the discussion considering the comments.

In this study, we verified the genetic difference between metastatic and non-metastatic behavior in breast cancer cells with RNA-sequencing data analysis; tried to understand the role of hub gene set in breast cancer metastasis; estimated the accuracy of prediction model with various algorithms; and compared the data with another breast cancer metastasis dataset.

We have proved the first finding with RNA-seq analysis data set. Single-cell RNA-sequencing data set from the GEO database included 94 single-cell samples from one patient. The metastatic samples were cancer cells from the lymph nodes and the non-metastatic samples were from breast cancer. We analyzed the difference between the two groups by PCA (Fig 1). However, the two groups were not clearly separated. In PC1 and PC2, some samples were mixed, and the heatmap revealed no large differences between these groups. Isolated tumor tissues from breast and lymph node consist of multiple types of cancer cells, including healthy cells [35]. The metastatic cells had a genetic profile similar to confirmed breast cancer tissue [36]. The PCA results revealed a slight separation between metastatic and non-metastatic cells. We identified the 2,683 genes (among 64,600 genes in total) related to metastasis and further analyzed them [17]. Differential gene expression was visualized in a volcano plot (Fig 1). The 2,683 genes were too much for functional assay, hence, we decided to separate them into gene modules by gene expression pattern. Each module presents detailed genomic functions and makes it easier to analyze and define key genes that are different in the metastatic and the non-metastatic groups. Gene functions were analyzed by WGCNA package which revealed clustered modules of genes showing similar expression to identify related genes in network analysis [37]. We first clustered the samples (Fig 2) and calculated the soft threshold power when β = 9 (Fig 3), which revealed 8 modules. Among them, the brown module showed the strongest relationship with metastasis (R = 0.46, p-value = 3 × 10-0.6) (Fig 4). We performed the functional enrichment assay with the genes from the brown module using the DAVID web tool and received many GO and KEGG terms. Most of the functions of the brown module genes were related to proliferation such as cell division, cell cycle, and chromosome separation, and were expressed in cytosol and nucleus where cell proliferation occurred. In KEGG pathway, terms of cell cycle and meiosis were identified. In this analysis, we found that metastatic cells have more cell proliferation activity, however it is difficult to know specifically which genes have key role in the function.

Next, we tried to determine the role of hub gene set in cancer metastasis by WGCNA assay. WGCNA analysis is a method of classifying genes with similar expression patterns into modules. Each module has an index for its degree of relevance. It is likely that genes belonging to modules with high relevance are involved in the same function in the cell. Genes from a module that are involved in key functions are defined as hub genes. We found 3 hub genes related to the cancer metastasis, involved to the mitochondrial metabolism. Mitochondrial metabolism genes are well known to be related to breast cancer metastasis in many studies.

We estimated the accuracy of prediction model with various algorithms to determine the most suitable machine learning algorithm for breast cancer metastasis. Total gene, the functional module, and hub genes were used as inputs to evaluate the accuracy of the machine learning prediction model. Among the algorithms used for estimation, RF was found to be the most suitable for breast cancer metastasis due to its high accuracy in predicting breast cancer cells. Variations in accuracy were also observed depending on the dataset used as input. The accuracy of the prediction model using the total gene was low and the overall accuracy was higher with functional module and the highest with the hub genes.

Finally, we compared RNA-seq data with another breast cancer metastasis dataset. The limitation of this study is the single-cell RNA-sequencing data from one person. It is impossible to generalize the findings of this study. To solve this problem, we compared our RNA-seq data with another breast cancer dataset. The data set of breast cancer patients taken from TCGA was classified into the M0 or M1 group with reference to clinical data. The accuracy of the predictive model was verified by the expression of genes corresponding to the three hub genes that we believed to be related to breast cancer. As a result, it was found that the external validation was 91% accurate. This confirms that the prediction model we built has high accuracy even with other datasets.

The limitation of this study should be discussed. I think this study needs external validation using real patient data or other database data. because single cell RNA-seq data from 'only one' patient have been used in this study. It is impossible to generalize the findings of this study.

The main limitation of this study has been mentioned in the discussion, and the results of external validation have been added to overcome the limitation.

Methods section: For external validation, RNA-seq data of 96 breast cancer samples, from The Cancer Genome Atlas (TGCA), were used. This data set consisted of 72 M0 macrophage non-metastatic and 24 M1 macrophage metastatic samples. To validate the prediction model, genes from hub gene set was used in external validation with RF. 

Result section: The train accuracy and test accuracy were both found to be 0.910, by external validation with breast cancer patient data from TCGA and RF.

Round 2

Reviewer 1 Report

This paper focuses on the machine learning model for breast cancer metastasis prediction by using random forest algorithm and mitochondrial metabolism hub gene assay.

Line 50 – put the capital I in the beginning of the sentence - it is possible…

Line 51 – put the t – letter in front of the word – he RAS

Line 109 – put the capital s in the beginning of the Pearson’s correlation coefficient – sij

Line 144 – put the capital G in the beginning of the word –gene set….

Line 159 – put the capital R at the following word – 3.1. result

Line 172 – put the capital in front of the text – the top…

Line 218 – Figure 1 a) and b) are not so clear, please increase the resolution.

Line 223 – Figure 3 a), b) and c) are not so clear, please increase the resolution.

Line 226 – Figure 4 a), b), c) and d) are not so clear, please increase the resolution.

Line 240 – remove one m from the word – metabolism

Line 268 – put number 3 instead of number 4  at the following text – Fig 3

Line 279 – put the capital N in front of the word Nor-

Line 282 – put the capital C in front of the word Cell

Line 283 – remove a full stop

The structure of the - testing various machine learning algorithms is missing. Please do the design of it.

More diagrams should be included in the paper, to have clear picture about the obtained results, for example 3.1.3. GO and pathway enrichment analysis is missing such a diagram.

Author Response

Point-by-point responses to reviewers’ comments

Reviewer 1

The structure of the - testing various machine learning algorithms is missing. Please do the design of it.

Response: We added the test accuracy information on various machine learning methods to the revised version of Table 2.

More diagrams should be included in the paper, to have clear picture about the obtained results, for example 3.1.3. GO and pathway enrichment analysis is missing such a diagram.

Response: Thank you for your comment. Accordingly, we have changed the GO term diagram in the revised Figure 4.

Reviewer 2 Report

I certainly found there have been progress but still want some more things to be improved.

1) Abstract

Machine-learning models that successfully predict breast cancer behavior and prognosis using genomic data are being developed. Researchers are trying to find gene mutations that can be used to predict metastasis in breast cancer using whole-genome or transcriptome data. However, it is difficult to determine the best metastatic marker genes among thousands of differentially expressed genes.

I suggest to summarize the first and second sentences into one sentence.

It is unclear why it is hard to determine the best metastatic genes by this sentence alone. Just because of magnitude of data size? It should be emphasized what is the biggest obstacle/problem here. And it should be logically linked to the purpose and design of this study which should be presented in the following sentence. The second and the third sentence here are not logically linked in current manuscript. You need to make readers understand what is the problem and how you solved this problem in this study.

and this should be the main difference and strong point of your study among other studies.

The method and results of the external validation part using TCGA is poorly described in the abstract.

Introduction

Generally it is well done reorganizing the introduction into three paragraphs.

However, it is better to use the topic sentences for the first sentences of each paragraph. Topic sentences mean the main sentences that summarize the all content of each paragraph. And this should be corresponded with the sentences in the abstract. For example, the first sentence of abstract corresponds the first sentence of first paragraph of introduction. The second sentence of abstract should corresponds the first sentence of second paragraph of the introduction. This is called structure of scientific writing. 

Still there is not enough logical reasoning what is the main problem/issue in background of this study, and how authors are trying to solve this problem with a certain/novel way of methods and study design in this study. Which does not attract readers and make readers lose interest to read this article.

Material and Methods

IRB should be waived by IRB and waiver number should be clearly mentioned even if this study uses only publicly opened dataset.

Still there are many sentences that should be moved to Results.

You need to describe only methods in the methods.

Results (Is the subheading Results or Result? Please check on this once again)

Still there are too many methods parts in the results that look redundant.

Please move all the sentence related to methods to methods parts.

This is very important thing to be revised before publication.

Figures and Tables

why did you put together all the figures and tables under one subheadings?

I assumed that figures and tables should be appropriately located among the manuscripts when it appear.

Please check on this again and reconsider rearrange them along the text.

Many of texts in the figures are not recognizable. Please check all the figures one by one and adjust the size of fonts especially after it is embedded in the manuscript file.

Discussion

There have been much improvement. However, it is still not enough for publication.

In the first paragraph. You mentioned what you have done in this study (which is techinically methods not discussion).

You should describe the results and the most important findings but not what you have done in this study. I think authors are still confusing about this.

And these findings should be logically corresponded to the questions and hypothesis that authors raised in the introduction.

How can you prove your design or your analysis using openly public data is superior or better than other studies or previous analysis done by other researchers.

if this study is the first study using this analysis, then it is encouraged to mention it. if you have very unique methodology or date processing, you need to emphasize this. 

And the purpose of discussion is to compare my results with the results of other researchers and re-evaluate your work in the context of other researches. 

In this discussion, authors are repeating the findings of their study. but no real discussion about their study considering other techniques or other researcher's analysis.

Conclusion

Conclusion is a summary of your findings and the overall meaning but not abstract. Current form is rather abstract that describes what authors have done in this study. Please rewrite conclusion.

Funding

I dont know why authors mentioned their funder in the acknowledgments but they mentioned there is no external funding in funding part.

I also think it is inappropriate to mention the English editing company for acknowlegments. There are several missed dots at line 341, line 347. 

Please check the errors in author contributions at line 338-340.

Author Response

Point-by-point responses to reviewers’ comments

Reviewer 2

1) Abstract

Machine-learning models that successfully predict breast cancer behavior and prognosis using genomic data are being developed. Researchers are trying to find gene mutations that can be used to predict metastasis in breast cancer using whole-genome or transcriptome data. However, it is difficult to determine the best metastatic marker genes among thousands of` differentially expressed genes.

I suggest to summarize the first and second sentences into one sentence.

It is unclear why it is hard to determine the best metastatic genes by this sentence alone. Just because of magnitude of data size? It should be emphasized what is the biggest obstacle/problem here. And it should be logically linked to the purpose and design of this study which should be presented in the following sentence. The second and the third sentence here are not logically linked in current manuscript. You need to make readers understand what is the problem and how you solved this problem in this study.

and this should be the main difference and strong point of your study among other studies.

Response: Thank you for your comment. We agree with you; hence, we have revised the abstract accordingly as follows:

“Breast cancer metastasis can have a fatal outcome, with the prediction of metastasis being critical for establishing effective treatment strategies. RNA-sequencing (RNA-seq) is a good tool for identifying genes that promote and support metastasis development; however, such information commonly explains the current situation of breast cancer while it fails to predict metastasis. Therefore, it is necessary to find a method that can predict breast cancer metastasis using RNA-seq results.” (page 1, lines 14–18)

The method and results of the external validation part using TCGA is poorly described in the abstract.

Response: Thank you for your comment and accept your kind advise. To correct this, we added the description about the external validation in the revised abstract as follows

“A breast cancer metastasis dataset from The Cancer Genome Atlas was used for external validation, showing an accuracy of over 91%.” (page 1, lines 28–29)

Introduction

Generally it is well done reorganizing the introduction into three paragraphs.

However, it is better to use the topic sentences for the first sentences of each paragraph. Topic sentences mean the main sentences that summarize the all content of each paragraph. And this should be corresponded with the sentences in the abstract. For example, the first sentence of abstract corresponds the first sentence of first paragraph of introduction. The second sentence of abstract should corresponds the first sentence of second paragraph of the introduction. This is called structure of scientific writing.

Still there is not enough logical reasoning what is the main problem/issue in background of this study, and how authors are trying to solve this problem with a certain/novel way of methods and study design in this study. Which does not attract readers and make readers lose interest to read this article.

Response:  Thank you for the comment and kind advice. The introduction was rewritten as follows:

“RNA-sequencing (RNA-seq) is being used to diagnose cancer and predict the behavior of cancer cells [1], which is directly linked to the expression of certain genes. Thus, it is possible to diagnose breast cancer and predict metastasis by analyzing gene expression profiles [2]. Genes involved in metastasis can be identified by comparing RNA-seq results of confirmed metastatic and non-metastatic breast cancer samples. Genes such as SETDB1 [3], MALAT1 [4], EHMT2 [5], RAB11B-AS1 [6], STAT3 [7], and RAS [8] were identified to play a role in lymph node metastasis of breast cancer. However, it is still impossible to effectively predict lymph node metastasis of breast cancer solely through gene expression analysis, although several studies have explored these particular genes. This limitation is because RNA-seq results only indicate the current state of breast cancer cells. Nevertheless, it is possible predict the behavior of breast cancer cells to some extent by analyzing the expression status of genes related to metastasis and grasping the current metastasis status of breast cancer [9], but the accuracy of such assessment is not high enough for clinical application.

Machine learning is a field of artificial intelligence that learns using algorithms based on real data, which is then used to design predictive models [11]. Such models are being actively investigated in different fields, including for the prediction of cancer prognosis using nuclear imaging [10]. However, prediction models build on nuclear imaging data still lack accuracy, making it difficult to apply clinically. This poor response of prediction models based only on imaging data can be explained as several factors can contribute for the cancer progression pattern. Recently, a prediction model with high accuracy was created by Dai et al. using genomic sequencing data of cancer cells that have metastasized from the colorectal cancer site to the lymph nodes [12]. This previous report demonstrates that it is possible to use next generation sequencing (NGS) data as input of prediction models with high accuracy.

In this study, a machine learning model based on RNA-seq data was developed using the random forest algorithm and hub genes to estimate the accuracy of the breast cancer metastasis prediction model. The accuracy of the prediction model with total genes, functional group genes, and hub genes was compared using various methods to identify a suitable machine learning algorithm for prediction of breast cancer metastasis.” (pages 1–2, lines 35–64)

Material and Methods

IRB should be waived by IRB and waiver number should be clearly mentioned even if this study uses only publicly opened dataset.

Response: Thank you for comment. We have added this missing information to the revised version of the manuscript (IRB waiver number: kirams 2021-02-003). (page 14, lines 364–365)

Still there are many sentences that should be moved to Results.

Response: In light of your comment, we have revised the sentence “Low-quality reads (quality under 20) and short sequences (<35 base pairs) were trimmed by fastQC (v.1.1.0).” to “Quality check was performed by fastQC (v.1.1.0).” (page 2, line 76)

We have also deleted the number of genes “16,482 genes, 54 genes, and 3 genes” from the Methods section 2.6.

You need to describe only methods in the methods.

Response:  Thank you for the comment. Accordingly, the following sentences were removed from the Results:

“In this study, 94 single-cell samples from one breast cancer patient were used. Through RNA-seq analysis, 38 samples were defined as non-metastatic, and 56 samples as metastatic. These data were taken from the GEO database, which was developed by a previous breast cancer study that defined the cancer cell detected from lymph node of patients as metastatic cell, and the one detected from breast cancer as the non-metastatic one [31].” (revised Results, section 3.1.1; page 4, lines 157-162)

Results (Is the subheading Results or Result? Please check on this once again)

Response: We changed “Result” to “Results”.

Still there are too many methods parts in the results that look redundant.

Please move all the sentence related to methods to methods parts.

This is very important thing to be revised before publication.

Response: Accordingly, we deleted the sentences in results:

“After analysis of differentially expressed genes and gene selection as previously described, we identified 2,683 genes showing significant differences between metastatic and non-metastatic samples with functions related to metastasis. “(revised Results, section 3.1.1; page 4, lines 166-168)

“The average linkage method and Pearson’s correlation method were used to cluster the results of obtained from single breast cancer cells” (revised Results, section 3.1.2; page 4, lines 174-175)

Figures and Tables

why did you put together all the figures and tables under one subheadings?

I assumed that figures and tables should be appropriately located among the manuscripts when it appear.

Please check on this again and reconsider rearrange them along the text.

Response: Thank you for kind advise, Accordingly, we have changed the figures location into the Result section.

Many of texts in the figures are not recognizable. Please check all the figures one by one and adjust the size of fonts especially after it is embedded in the manuscript file.

Response: Thank you for the comment. Accordingly, we have changed the figures’ resolution and text font size to improve their readability.

There have been much improvement. However, it is still not enough for publication.

In the first paragraph. You mentioned what you have done in this study (which is techinically methods not discussion).

You should describe the results and the most important findings but not what you have done in this study. I think authors are still confusing about this.

And these findings should be logically corresponded to the questions and hypothesis that authors raised in the introduction.

How can you prove your design or your analysis using openly public data is superior or better than other studies or previous analysis done by other researchers.

if this study is the first study using this analysis, then it is encouraged to mention it. if you have very unique methodology or date processing, you need to emphasize this. 

And the purpose of discussion is to compare my results with the results of other researchers and re-evaluate your work in the context of other researches. 

In this discussion, authors are repeating the findings of their study. but no real discussion about their study considering other techniques or other researcher's analysis.

Response: Thank you for the comments and we accept your kind advise. Therefore, we have accordingly revised the Discussion section as follows:

“In this study, we used RNA-sequencing techniques and machine learning algorithms to build a model to predict breast cancer metastasis, which is promoted by genetic changes within cancer cells. Such a model can be used as the basis for diagnosis and treatment of breast cancer by analyzing changes in genetic events. In the past, only a limited number of genes could be analyzed to profile cancer cells, but advances in technology have enhanced the analytical methods, allowing the assessment of the genetic landscape of cells at once. RNA-seq is believed to hold the power to explain the behavior of cancer by identifying key genes. However, the behavior of cancer cells is difficult to explain based solely on the gene expression of one gene. To overcome this limitation, herein we used the concept of a functional group gene and a hub gene. The expression of a hub gene can accurately determine the function of a cell rather than using a single gene since it is closely related to the function of the cell. Several genes were found to be involved when the cancer cell function was addressed. Hence, genes involved in the same function were defined as the functional group genes, among which genes with the most essential functions were selected and defined as the hub genes.

WGCNA, a bioinformatics analysis method, was used to identify the functional group gene and the hub gene [32]. This approach allows to evaluate the expression pattern of a gene and construct a module of genes with similar expression patterns, which is generally used for gene analysis of cancer cells [33]. This method has the advantage of being able to search for a function related to a highly expressed gene among the functions of a cell, as it analyzes only the expression pattern of the gene. Gene modules were defined as functional groups, and hub genes were retrieved from within. The hub gene was selected as the gene with the most connections after confirming the association between the genes in the analyzed module using CHAT app in Cytoscape [34]. Therefore, the identified functional group and three hub genes could represent a more detailed and reliable indicator of lymph node metastasis in breast cancer.

A detailed analysis of the functional group genes revealed that they were mainly involved in cell division and proliferation (Fig. 5). Epithelial mesenchymal transition (EMT) is the most frequently expressed function in tumor metastases, and cell division and proliferation are active and essential actions during EMT and cancer cells spreading [35]. The identified hub genes (NDUFB3, NDUFB5, and NDUFA9) are mitochondrial genes that have various functions, but are known to be involved in mitochondria metabolism. This function is known to play an important role in metastasis of cancer cells and is reported to be overexpressed when metastasis of breast, rectal, and head and neck cancers occurs [36-38].

We used a machine learning method to construct a highly accurate predictive model for breast cancer metastasis. Several algorithms could be used for the machine learning process; hence, since it was not known which algorithm was suitable for breast cancer metastasis, we tested various algorithms. Among the algorithms used to estimate the predictive model, RF and GB showed the highest accuracy. These two most suitable algorithms revealed the highest degree of analysis because the number of analyzed samples was small, and a prediction model was constructed by extracting the most suitable few factors for analysis [39,40]. The sample used to design the predictive model was not a general result since it was based on single cell RNA-seq data. Therefore, a large-scale breast cancer metastasis TCGA RNA-seq dataset was used for external validation, which was classified into the M0 or M1 group with reference to the clinical data. The breast cancer metastasis model using RF showed a high accuracy of 94%, and the external validation also showed a high accuracy of 91%. (pages 13–14, lines 253–309)

Conclusion

Conclusion is a summary of your findings and the overall meaning but not abstract. Current form is rather abstract that describes what authors have done in this study. Please rewrite conclusion.

Response: Thank you for the comment. Accordingly, we have revised the Conclusion as follows:

“In this study, we applied RNA-seq data and machine learning algorithms for the first time to construct a highly accurate predictive model to predict breast cancer metastasis. The field of predicting patient prognosis using machine learning has been mainly applied to nuclear imaging. Although the results of analyzing imaging factors, such as computerized tomography and magnetic resonance imaging, were applied to machine learning to estimate the accuracy of the predictive model, it is still difficult to apply such approaches in the clinical setting. Genetic data obtained by RNA sequencing can describe the patient condition more accurately than data obtained by image analysis. Therefore, we expect that the novel predictive model may pave the way for enhance prognosis assessment of patients with breast cancer. Also, as we combined RNA-seq data with machine learning to estimate predictive models with high levels of accuracy, applying nuclear imaging and RNA-seq to predictive models at the same time will allows us estimate better predictive models. It is expected that the predictive model estimated through this combined approach may be used in clinical practice.” (pages 14–15, lines 346–359)

Funding

I don’t know why authors mentioned their funder in the acknowledgments but they mentioned there is no external funding in funding part.

Response: Thank you for this comment. We have corrected this inconsistency by moving the funding information to its correct place.

I also think it is inappropriate to mention the English editing company for acknowlegments.

Response: Thank you for your kind advise. Accordingly, we deleted the English editing company information from acknowledgment.

There are several missed dots at line 341, line 347. 

Response: Thank you comment. We have revised this information accordingly, changed follow:

Line 341 was changed to “This study was supported by the National Research Foundation of Korea (NRF) grant funded by the Korea government (Ministry of Science and ICT) (No. 2020M2D9A1094070).”

Line 347 was added dot as follow: “Conflicts of Interest: The authors declare no conflict of interest.”

Please check the errors in author contributions at line 338-340.

Response: Thank you comment. We have corrected errors as follow:

“Byung-Chul Kim; conceptualization, Byung-Chul Kim and Sang-Keun Woo; methodology, Byung-Chul Kim; software, Byung-Chul Kim; validation, Byung-Chul Kim and Jingyu Kim; formal analysis, Byung-Chul Kim; investigation, Byung-Chul Kim; data curation, Byung-Chul Kim; writing—original draft preparation, Byung-Chul Kim; writing—review and editing, Ilhan Lim, Dong Ho Kim, Sang Moo Lim and Sang-Keun Woo; visualization, Byung-Chul Kim; super-vision, Sang-Keun Woo; project administration, Ilhan Lim. All authors have read and agreed to the published version of the manuscript.”

Round 3

Reviewer 2 Report

I think there have been a big progress. Some more refinement is required.

As I understand, the authors adopted a concept of using "hub gene" instead of analyzing individual genes from RNA-seq data, that are more representative for functional characteristics of genetic change in cancer cells. This is the special thing authors adopted in this study design.

However, it is not clearly mentioned and not emphasized in the manuscript especially in introduction. (The first part of discussion is good explanation why authors adopted this in this study.) It is desirable to mention why and how you adopted "hub gene" analysis in introduction. Also, in abstract, authors mentioned about this as below.

"however, such information commonly explains the current situation of breast cancer while it fails to predict metastasis. Therefore, it is necessary to find a method that can predict breast cancer metastasis using RNA-seq results."

 But it is a bit not enough to understand why authors use "hub gene anaylsis". Please reword these sentences to emphasize the specialty of this study and make readers to understand the relevancy of authors hypothesis.

1) Introduction

It is generally well revised. However, as I mentioned earlier, there should be more explanation about "hub gene" analysis which is essential of this study. why it is needed, how authors reached this idea etc. Maybe the first part of discussion can be briefly mentioned here.

2) Please check all the figures once again and make them easily recognizable.

(1) I guess it will be better for readers to understand if you make a diagram of methods explaining about the each step of analysis (how you find the hub genes). Please make this kind of diagram and add it as a figure in methods.

(2) Fig 1a & b - impossible to recognize the axis letters and the dots in the figures - please make them recognizable.

(3) Fig 4 a, b, c - the title of each figure is too small. Please make them larger enough to be easily recognized. The letters of y axis are too small as well.

4) Discussion

Please mention the major findings of this study at the first paragraph and it is better to explain about in-depth explanation about hub genes and other stuffs after that.

Author Response

Point-by-point responses to reviewers’ comments

Reviewer 2

I think there have been a big progress. Some more refinement is required.

As I understand, the authors adopted a concept of using "hub gene" instead of analyzing individual genes from RNA-seq data, that are more representative for functional characteristics of genetic change in cancer cells. This is the special thing authors adopted in this study design.

However, it is not clearly mentioned and not emphasized in the manuscript especially in introduction. (The first part of discussion is good explanation why authors adopted this in this study.) It is desirable to mention why and how you adopted "hub gene" analysis in introduction. Also, in abstract, authors mentioned about this as below.

"however, such information commonly explains the current situation of breast cancer while it fails to predict metastasis. Therefore, it is necessary to find a method that can predict breast cancer metastasis using RNA-seq results."

 But it is a bit not enough to understand why authors use "hub gene anaylsis". Please reword these sentences to emphasize the specialty of this study and make readers to understand the relevancy of authors hypothesis.

Response: Thank you for your comment. We agree with you; hence, we have revised the abstract accordingly as follows:

The hub gene analysis method is a bioinformatics method that can effectively analyze RNA sequencing results. This can be used to specify the set of genes most relevant to the function of the cell involved in metastasis.” (page 1, lines 16–18)

 1) Introduction

It is generally well revised. However, as I mentioned earlier, there should be more explanation about "hub gene" analysis which is essential of this study. why it is needed, how authors reached this idea etc. Maybe the first part of discussion can be briefly mentioned here.

Response: Thank you for your comment and accept your kind advise. We have added this sentence to the revised version of the manuscript as follows:

“Hub gene was used to obtain more accuracy for represent the function of cells than single gene analysis from RNS-sequencing in the cancer studies [10]. Hub gene was defined as genes have top 10% of connectivity in individual gene module related gene function [11]. The gene modules were created by a systematic biological strategy for evaluating gene association patterns among different samples with bioinformatics tools like WGCNA or GSEA [12].” (page 2, lines 48–53)

(1) I guess it will be better for readers to understand if you make a diagram of methods explaining about the each step of analysis (how you find the hub genes). Please make this kind of diagram and add it as a figure in methods.

Response: Thank you for your comment and accept your kind advise.  We have added new diagram for understanding the process of hub gene set specification.

(2) Fig 1a & b - impossible to recognize the axis letters and the dots in the figures - please make them recognizable.

Response: Thank you for your comment and accept your kind advise.  We have changed the size of the axis letters and the dots in figures.

(3) Fig 4 a, b, c - the title of each figure is too small. Please make them larger enough to be easily recognized. The letters of y axis are too small as well.

Response: Thank you for your comment and accept your kind advise.  We have changed the size of the title of each figure and y axis letters.

4) Discussion

Please mention the major findings of this study at the first paragraph and it is better to explain about in-depth explanation about hub genes and other stuffs after that.

Response: Thank you for your comment and accept your kind advise. We have revised the discussion as follows:

“In this study, we applied hub gene set from RNA-seq and machine learning algorithms for the first time to construct predictive models for breast cancer metastasis with high accuracy, contrasting with most of the previous studies that have focused on nuclear imaging data. Although the results of analyzing imaging factors, such as computerized tomography and magnetic resonance imaging were applied to machine learning to estimate the accuracy of the predictive model, it is still difficult to apply in the clinical setting. Genetic data obtained by next generation sequencing can describe the patient clinical condition more accurately than data obtained solely by image analysis. Therefore, we expected that in the construction of this prediction model, higher accuracy than the existing image analysis-based prediction model could be estimated.

RNA-sequencing techniques and machine learning algorithms was used to build a model to predict breast cancer metastasis, which is promoted by genetic changes within cancer cells. Such a model can be used as the basis for diagnosis and treatment of breast cancer by analyzing changes in genetic events. In the past, only a limited number of genes could be analyzed to profile cancer cells, but advances in technology have enhanced the analytical methods, allowing the assessment of the genetic landscape of cells at once. RNA-seq is believed to hold the power to explain the behavior of cancer by identifying key genes. However, the behavior of cancer cells is difficult to explain based solely on the gene expression of one gene. To overcome this limitation, herein we used the concept of a functional group gene and a hub gene. The expression of a hub gene can accurately determine the function of a cell rather than using a single gene since it is closely related to the function of the cell. Several genes were found to be involved when the cancer cell function was addressed. Hence, genes involved in the same function were defined as the functional group genes, among which genes with the most essential functions were selected and defined as the hub genes.

WGCNA, a bioinformatics analysis method, was used to identify the functional group gene and the hub gene [35]. This approach allows to evaluate the expression pattern of a gene and construct a module of genes with similar expression patterns, which is generally used for gene analysis of cancer cells [36]. This method has the advantage of being able to search for a function related to a highly expressed gene among the functions of a cell, as it analyzes only the expression pattern of the gene. Gene modules were defined as functional groups, and hub genes were retrieved from within. The hub gene was selected as the gene with the most connections after confirming the association between the genes in the analyzed module using CHAT app in Cytoscape [37]. Therefore, the identified functional group and three hub genes could represent a more detailed and reliable indicator of lymph node metastasis in breast cancer.

A detailed analysis of the functional group genes revealed that they were mainly involved in cell division and proliferation (Fig. 6). Epithelial mesenchymal transition (EMT) is the most frequently expressed function in tumor metastases, and cell division and proliferation are active and essential actions during EMT and cancer cells spreading [38]. The identified hub genes (NDUFB3, NDUFB5, and NDUFA9) are mitochondrial genes that have various functions, but are known to be involved in mitochondria metabolism. This function is known to play an important role in metastasis of cancer cells and is reported to be overexpressed when metastasis of breast, rectal, and head and neck cancers occurs [39-41].

We used a machine learning method to construct a highly accurate predictive model for breast cancer metastasis. Several algorithms could be used for the machine learning process; hence, since it was not known which algorithm was suitable for breast cancer metastasis, we tested various algorithms. Among the algorithms used to estimate the predictive model, RF and GB showed the highest accuracy. These two most suitable algorithms revealed the highest degree of analysis because the number of analyzed samples was small, and a prediction model was constructed by extracting the most suitable few factors for analysis [42,43]. The sample used to design the predictive model was not a general result since it was based on single cell RNA-seq data. Therefore, a large-scale breast cancer metastasis TCGA RNA-seq dataset was used for external validation, which was classified into the M0 or M1 group with reference to the clinical data. The breast cancer metastasis model using RF showed a high accuracy of 94%, and the external validation also showed a high accuracy of 91%.” (page 10, line 261-270)